# Differences in Dietary Intakes among Lebanese Adults over a Decade: Results from Two National Surveys 1997–2008/2009

**Lara Nasreddine [1], Jennifer J. Ayoub [1], Fatima Hachem [2], Jiana Tabbara [3], Abla M. Sibai [3], Nahla Hwalla [1]**  **and Farah Naja [1],***

[1] Nutrition and Food Sciences Department, Faculty of Agriculture and Food Sciences, American University of Beirut, P.O. Box 11-0236, Riad El Solh, Beirut 11072020, Lebanon
[2] Nutrition and Food Systems Division, Food and Agricultural Organization of the United Nations, 00153 Rome, Italy
[3] Epidemiology and Population Health Department, Faculty of Health Sciences, American University of Beirut, P.O. Box 11-0236, Riad El Solh, Beirut 11072020, Lebanon
* Correspondence: fn14@aub.edu.lb; Tel.: +961-1-350-000 (ext. 4504)

**Abstract:** Amidst the ongoing societal and economic shifts in the Eastern Mediterranean region (EMR), this study aims at investigating temporal trends in food consumption and nutrient intakes among Lebanese adults, by age and sex. Data were derived from two national cross-sectional surveys conducted in Lebanon during two time periods (1997; 2008/2009). In both surveys, dietary assessment was based on 24-h recalls. The results, expressed as % energy intake (%EI), revealed a significant decrease ($p < 0.001$) in the consumption of bread, fruits, fresh fruit juices, milk and eggs, whereas the consumption of added fats and oils, poultry, cereals and cereal-based products, chips and salty crackers, sweetened milk and hot beverages increased over time ($p < 0.001$). A significant increase in dietary energy (kcal/day) and fat intake (%EI) was observed, coupled with decreases in carbohydrate intake (%EI) and dietary density of vitamin A and vitamin C (per 1000 kcal) ($p < 0.001$). These changes were noted in both genders and across age groups, albeit there were some disparities between groups. In conclusion, based on national nutrition surveys, this study is the first to characterize the nutrition transition in a middle-income country of the EMR, shedding light on priority areas for nutrition policies and interventions.

**Keywords:** nutrition transition; Eastern Mediterranean region; adults; dietary intakes; nutrition surveys; Lebanon

## 1. Introduction

Suboptimal diet is recognized as a major cause of poor health and a risk factor for several non-communicable diseases (NCDs), including diabetes, cardiovascular diseases and certain types of cancer [1]. In the Eastern Mediterranean region (EMR), suboptimal diet was reported as the leading risk factor for cardiometabolic disease (CMD) mortality, contributing to 435,174 deaths in 2010, which represented 48% of total CMD [1]. The sociocultural changes, including modernization, urbanization and economic development that most countries of the region have witnessed during the past decades may have led to increased availability, affordability and consumption of unhealthy foods, and hence to changes in the population's food consumption patterns and dietary habits [2].

Available evidence, which is based on food supply data from food balance sheets (FBSs), suggests that countries in the EMR are undergoing a nutrition transition with its characteristic shifts in diet, towards a higher supply of energy, fat and sugar [2]. This type of evidence is however constrained

by the inherent limitations of FBS data as these give only a crude, and over-estimated value of foods available for consumption per head of the population, irrespective of age or gender [3]. FBS data are also limited by the fact that they do not take into account food and nutrient losses that may occur as a result of processing, spoilage, trimming or waste [3]. To overcome these limitations, numerous countries resorted to individual nutrition surveys for the examination of dietary shifts and food consumption trends at the population level [4–7]. In contrast to FBSs, data from individual surveys more closely reflect actual food consumption and provide information not only on the average intake levels, but also on their distribution across various groups within the population [3]. The investigation of temporal dietary trends based on individual surveys would thus allow for the assessment of changes in the population's food consumption and nutrient intakes, provide insight on how these trends may vary between population subgroups and permit the identification of priority areas for intervention in the country [6].

Studies investigating trends in food consumption based on individual nutrition surveys are lacking in the EMR. An important prerequisite for such studies is the availability of nationally representative food consumption surveys conducted at different points in time. In Lebanon, a small country in the Eastern Mediterranean basin, two nationally representative individual food consumption surveys were conducted in 1997 and 2008/2009, respectively, providing data on the consumption level of various food groups, as well as the intake of macro- and micronutrients in the adult Lebanese population [2,8]. During the past decade, Lebanon has witnessed increased urbanization, economic development and modernization in lifestyle, all of which may have contributed to changes in the population's food consumption patterns. In fact, Lebanon is considered as one of the most urbanized countries in the world, with 88% of its population of approximately 4 million residing in urban areas [9]. Lebanon has also undergone significant improvements in its economic indicators, with its gross domestic product (GDP) increasing from 15.75 to 35.47 billion US dollars (USD) from 1997 to 2009 and per capita gross national income (GNI) increasing from 4730 to 7920 USD during the same period of time [10]. Changes in food consumption patterns may have been further accentuated in Lebanon by the lack of national policy to promote and sustain production and consumption of foods characteristic to the country [11] and by the continuous and accrued social instability and political turmoil. Amidst these ongoing societal shifts, developments and challenges, this study aims to investigate temporal trends in food consumption and nutrient intake among Lebanese adults between 1997 and 2008/2009 and examine these trends in population subgroups based on age and sex. Findings of this study will contribute to the characterization of the nutrition transition in a middle-income country of the EMR.

## 2. Methods

### 2.1. Survey Designs

The data for the present analysis were derived from two national cross-sectional surveys conducted in Lebanon during two time periods, 1997 and 2008/2009. Details about the protocols used in these surveys are published elsewhere [12]. AMS and NH led the conceptualization and execution of both surveys, whereby similar protocols for sampling and data collection were used. In both surveys, households were the sampling units. The door-to-door approach was used to obtain nationally representative samples of Lebanese households. The distribution of participants in both surveys was proportional to that of the Lebanese population, according to age, sex and district distribution [13,14]. The protocols for these surveys were approved by the Institutional Review Board of the American University of Beirut. Appropriate consents/assents were obtained from subjects prior to participation.

### 2.2. Design of the 1997 National Survey

The sampling framework of the 1997 survey was based on the Population and Housing Survey (PHS) conducted by the Lebanese Ministry of Social Affairs and the United Nations Population Fund (UNFPA) [14]. For the PHS, the primary observation units were cities, towns or groups of towns having

small populations, called population conglomerations (*n* = 637). Each population conglomeration unit was then divided to 4284 sections or geographic enclaves (second-level observation units), of which 1422 were drawn at random. Within each enclave, 50 households were randomly selected. The final sample size of the PHS was 61,580 (response rate = 95.5%) [15]. For the purpose of the 1997 survey, a random sub-sample of 1% of the PHS households was selected [16]. The field work was conducted between April 1997 and September 1997, and the final sample consisted of a total of 2004 subjects (refusal rate 10%). Of those, adults who were aged 20 years or older and had complete data were included in this study (*n* = 1063).

*2.3. Design of the 2008/2009 National Survey*

The 2008/2009 survey included a nationally representative sample of individuals aged >6 years. The design and conduct of the survey are described in detail elsewhere [17]. For this year's survey, households were selected using a stratified cluster sampling, whereby the Lebanese governorates constituted the various strata while districts within governorates were considered clusters. Within clusters, households were selected at random using probability proportional to size sampling and considering the distribution of the Lebanese population (by sex and 5-year age group) estimated by the Central Administration for Statistics in Lebanon (2004) [13]. In each household, one adult and one child/adolescent were selected for participation, using a household roster. The field work was conducted between May 2008 and August 2009 with a final sample including 3636 subjects (refusal rate 10.7%). For the purpose of this study, data pertinent to adults participating in the 2008/2009 national survey and who had complete data were included (*n* = 2518).

*2.4. Data Collection Protocols*

For both surveys, nutritionists conducted data collection in the participants' house using face-to-face interviews, with each interview lasting for around 60 minutes. All nutritionists involved in data collection underwent extensive training in order to standardize interviewing techniques and minimize the interviewer's bias. The questionnaire used in the data collection consisted of four sections: (1) demographic, (2) socioeconomic, (3) medical history and (4) health-seeking behavior. Common variables between the questionnaires used in both surveys included age (in years), sex (males and females), governorates (Beirut, Mount Lebanon, North, South and Bekaa), marital status (single, married and divorced/separated/widowed), education level (primary education or less, up to high school education, and university degree or higher), employment status (employed and not employed) and crowding index. The latter is considered a proxy for socioeconomic status and is calculated by dividing the number of household members by the number of rooms used for sleeping as the denominator. Several epidemiological studies have correlated a high household crowding index with low socioeconomic status [18,19].

*2.5. Dietary Intake Assessment*

In addition to the multicomponent questionnaire, data collection in both surveys involved the collection of 24-h recalls (24-HRs) for the assessment of dietary intake. The 24-HRs were carried out using the multiple pass food recall (MPR) 5-step approach, developed by the United States Department of Agriculture (USDA) [20]. This approach has consistently showed attenuation in the 24-HRs' limitations [20,21]. The five steps followed included (1) quick food list recall, (2) forgotten food list probe, (3) time and occasion at which foods were consumed, (4) detailed overall cycle and (5) final probe review of the foods consumed. For each 24-HR, the interviewer obtained information regarding dietary intake during the past 24 h, related to the time of each meal's intake, the food consumed by the subject, its portion size, preparation methods, and the brand of the food and beverages consumed, if applicable. For the estimation of portion size, a reference portion, representing one standard serving expressed in household measures, was defined. Common household measures used were measuring cups, spoons and plates. For certain food items, portion size photos were also used.

Nutritionist Pro software (version 5.1.0, 2014, First Data Bank, Nutritionist Pro, Axxya Systems, San Bruno, CA, USA) was used for the analysis of the dietary intake data and to estimate energy, and macro- and micronutrients' intakes. For composite and mixed dishes, standardized recipes were added to the Nutritionist Pro software using single food items. Within the Nutritionist Pro, the USDA database was selected for analysis (SR 24, published September 2011).

### 2.6. Statistical Analysis

Proportions were used to represent demographic and socioeconomic characteristics of study participants for both surveys (1997 and 2008/2009). These characteristics were compared between the two years using chi-squared tests. Dietary intake data was presented for all adults as well as by age groups (20–39.9 years, 40–59.9 years and ≥60 years), as means ± standard error (SE). Except for energy and fiber, macronutrients, micronutrients and food groups' intakes were indicated as percent contribution to total energy. In order to adjust for differences in distributions between the two samples across socio-demographic variables, the analysis of covariance (ANCOVA) was used to estimate adjusted means of dietary intake. In this latter analysis, the dietary intake variable was the dependent variable, the year of data collection was the fixed factor and sex, governorates, marital status, education level and employment status were the independent covariates.

## 3. Results

### 3.1. Sociodemographic Characteristics of Participants

Table 1 shows the demographic and socioeconomic characteristics of participants aged ≥20 years, by survey year (1997 and 2008/2009). The study sample included 1063 adults in 1997 and 2518 adults in 2008/2009. While there was no difference according to crowding index, the distributions of age, sex, governorate, marital status, educational level and employment status were different between the two surveys. In 2008/2009, compared to 1997, greater proportions of participants were aged 20–39.9 years, were males and were single. On the other hand, the proportions of unemployed individuals and those with primary education level were lower in 2008/2009 compared to 1997 (Table 1).

**Table 1.** Sociodemographic characteristics of the total study population (*n* = 3581) by survey year, Lebanon, 1997 (*n* = 1063) and 2008/2009 (*n* = 2518).

| | 1997 (*n* = 1063) | 2008/2009 (*n* = 2518) | Total (*n* = 3581) | *p*-Value |
|---|---|---|---|---|
| | **Mean ± SD** | | | |
| **Age (years)** | 43.4 ± 15.5 | 41.6 ± 16.1 | 42.1 ± 15.9 | **0.002** |
| | ***n* (%)** | | | |
| **Age (years)** | | | | |
| 20–39.9 | 488 (45.9) | 1317 (52.3) | 1805 (50.4) | |
| 40–59.9 | 383 (36.0) | 780 (31.0) | 1163 (32.5) | **0.002** |
| ≥60 | 192(18.1) | 421 (16.7) | 613 (17.1) | |
| **Sex** | | | | |
| Males | 436 (41.0) | 1136 (45.1) | 1572 (43.9) | **0.024** |
| Females | 627 (59.0) | 1382 (54.9) | 2009 (56.1) | |
| **Governorate** | | | | |
| Beirut | 237 (22.3) | 281 (11.2) | 518 (14.5) | |
| Mount Lebanon | 285 (26.8) | 1097 (43.6) | 1382 (38.6) | |
| North | 166 (15.6) | 419 (16.6) | 585 (16.3) | **<0.001** |
| South | 188 (17.7) | 425 (16.8) | 613 (17.1) | |
| Bekaa | 187 (17.6) | 296 (11.8) | 483 (13.5) | |
| **Marital status** | | | | |
| Single | 226 (21.3) | 785 (31.2) | 1011 (28.2) | |
| Married | 771 (72.5) | 1522 (60.5) | 2293 (64.1) | **<0.001** |
| Divorced, separated, or widowed | 66 (6.2) | 209 (8.3) | 275 (7.7) | |

**Table 1.** *Cont.*

|  | 1997 (*n* = 1063) | 2008/2009 (*n* = 2518) | Total (*n* = 3581) | *p*-Value |
|---|---|---|---|---|
|  | Mean ± SD | | | |
| **Crowding index** | | | | |
| <1 Person/Room | 396 (37.3) | 951 (37.8) | 1347 (37.6) | 0.771 |
| ≥1 Person/Room | 667 (62.7) | 1567 (62.2) | 2234 (62.4) | |
| **Education level** | | | | |
| Primary or lower | 346 (32.5) | 531 (21.1) | 877 (24.5) | **<0.001** |
| Up to high school or technical school | 464 (43.7) | 1254 (49.8) | 1718 (48.0) | |
| University degree or higher | 253 (23.8) | 733 (29.1) | 986 (27.5) | |
| **Employment status** | | | | |
| Employed | 472 (44.4) | 1248 (49.6) | 1720 (48.0) | **0.001** |
| Not employed * | 591 (55.6) | 1269 (50.5) | 1860 (52.0) | |

Numbers in **bold** face are statistically significant (*p*-value ≤ 0.05). * Not employed includes individuals who are not working, do not want to work, looking for work or retired; housewives; students.

### 3.2. Food Groups (%EI)

Mean intake of various food groups (by age and sex) and their comparison between 1997 and 2008/2009 are presented in Table 2. These intakes are presented in terms of their contribution to total energy intake (%EI). Food items that were included in each of the food groups are described in Appendix A. Absolute intake of various food groups are presented in Supplementary Table S1.

In the overall study population, compared to 1997, in 2008/2009, intakes of bread, milk, eggs, fruits and fresh fruit juices decreased significantly (bread: 17.84 ± 0.31% vs. 21.81 ± 0.45%; milk: 1.09 ± 0.08% vs. 1.53 ± 0.11%; eggs: 0.74 ± 0.07% vs. 1.44 ± 0.1%; fruits: 4.72 ± 0.15% vs. 7.36 ± 0.22%; fresh fruit juices: 0.23 ± 0.04% vs. 0.42 ± 0.05%, in 2008/2009 and 1997, respectively). The differences in bread and fruit intakes were also observed among each of the age group categories. The decrease in milk intake was found to be significant only among participants aged 20–39.9 years. Eggs and fresh fruit juice intakes decreased among participants aged 20–39.9 years and 40–59.9 years. (Table 2)

In contrast, significant increases were observed in the percentage of energy intake of cereals and cereal-based products (11.45 ± 0.61% in 1997 vs. 15.66 ± 0.42% in 2008/2009) and added fats and oils (3.38 ± 0.21% in 1997 vs. 4.59 ± 0.15% in 2008/2009), in the overall study sample and across the different age groups. The significant increase in intakes of chips and salty crackers were observed in the overall study population (from 0.31 ± 0.1% in 1997 to 0.81 ± 0.07% in 2008/2009) and among participants aged 20–39.9 years. Furthermore, sweetened milk intake increased from 0.09 ± 0.06% to 0.26 ± 0.04%, and hot beverages from 0.37 ± 0.05% to 0.59 ± 0.04% in 1997 and 2008/2009, respectively (particularly among participants aged 20–39.9 and ≥60 years). Similar changes were shown in the percentage of energy intake from poultry and the miscellaneous food group, where the overall consumption increased from 2.68 ± 0.28% in 1997 to 3.95 ± 0.19% in 2008/2009 and from 0.58 ± 0.12% in 1997 to 1.12 ± 0.08% in 2008/2009, respectively, particularly among participants aged 20–39.9 years and 40–59.9 years. As for added sugars, jams, honey, and molasses, although no significant change was shown in their overall percentage of energy intake, a significant increase was observed among 40–59.9 years old from 1.11 ± 0.14% to 1.74 ± 0.1% in 1997 and 2008/2009, respectively. Moreover, there was a slightly significant increase in the percentage of energy intake from milk derivatives, among participants aged 40–59.9 years old, from 8.35 ± 0.48% in 1997 to 7.16 ± 0.36% in 2008/2009. Compared to 1997, in 2008/2009 no changes in the intake of legumes, starchy vegetables, vegetables, nuts and seeds, red meat, processed meat, fish, sweets, sugar sweetened beverages, alcoholic beverages and fast food were observed. (Table 2)

Of the differences in food groups' intake that were observed in the overall study population, the following were also observed for males and females: bread, cereal and cereal-based products, chips and salty crackers, milk, poultry, eggs, fruits, hot beverages, added fats and oils, and miscellaneous. The differences in sweetened milk were only shown among males, while those in fresh fruit juices were shown in females only (Table 2).

**Table 2.** Mean daily intake of various food groups (%EI) (by age and sex) and their comparison between 1997 and 2008/2009, in the study population (*n* = 3581), aged ≥20 years, Lebanon.

| | Males (n = 1572) | | | Females (n = 2009) | | | Both Sexes (n = 3581) | | |
|---|---|---|---|---|---|---|---|---|---|
| | 1997 (n = 436) | 2008/2009 (n = 1136) | *p*-Value | 1997 (n = 627) | 2008/2009 (n = 1382) | *p*-Value | 1997 (n = 1063) | 2008/2009 (n = 2518) | *p*-Value |
| **20–39.9 years** | | | | | | | | | |
| Bread | 23 ± 1.18 | 17.41 ± 0.66 | **<0.001** | 17.46 ± 0.77 | 14.78 ± 0.53 | **0.005** | 19.76 ± 0.65 | 15.81 ± 0.42 | **<0.001** |
| Cereals and Cereal-Based Products | 11.07 ± 1.53 | 17.18 ± 0.86 | **0.001** | 13.58 ± 1.13 | 15.25 ± 0.78 | 0.230 | 12.49 ± 0.91 | 16.21 ± 0.58 | **0.001** |
| Legumes | 3.48 ± 0.75 | 3.98 ± 0.42 | 0.569 | 2.01 ± 0.54 | 3.08 ± 0.37 | 0.110 | 2.67 ± 0.44 | 3.42 ± 0.28 | 0.158 |
| Starchy Vegetables | 4.66 ± 0.54 | 3.87 ± 0.3 | 0.207 | 4.66 ± 0.45 | 4.65 ± 0.31 | 0.977 | 4.62 ± 0.35 | 4.31 ± 0.22 | 0.461 |
| Vegetables | 6.11 ± 0.65 | 5.89 ± 0.36 | 0.770 | 9.32 ± 0.72 | 9.42 ± 0.5 | 0.908 | 7.95 ± 0.51 | 7.94 ± 0.32 | 0.983 |
| Chips and Salty Crackers | 0.16 ± 0.27 | 1.11 ± 0.15 | **0.003** | 0.63 ± 0.24 | 1.45 ± 0.16 | **0.006** | 0.47 ± 0.18 | 1.29 ± 0.11 | **<0.001** |
| Nuts and Seeds | 3.65 ± 0.6 | 1.97 ± 0.33 | **0.016** | 2.5 ± 0.46 | 2.73 ± 0.32 | 0.688 | 2.88 ± 0.36 | 2.39 ± 0.23 | 0.265 |
| Dairy Products | | | | | | | | | |
| Milk | 1.03 ± 0.21 | 0.78 ± 0.12 | 0.319 | 1.67 ± 0.22 | 1.2 ± 0.15 | 0.089 | 1.42 ± 0.16 | 1.02 ± 0.1 | **0.039** |
| Milk Derivatives | 5.98 ± 0.68 | 6.26 ± 0.38 | 0.719 | 6.83 ± 0.53 | 7.67 ± 0.37 | 0.199 | 6.5 ± 0.42 | 7.05 ± 0.26 | 0.275 |
| Sweetened Milk | 0 ± 0.12 | 0.23 ± 0.06 | 0.097 | 0.12 ± 0.11 | 0.28 ± 0.08 | 0.280 | 0.07 ± 0.08 | 0.25 ± 0.05 | 0.075 |
| Meat, Processed Meat, Poultry, Fish, Eggs | | | | | | | | | |
| Red Meat | 7.06 ± 0.91 | 6.84 ± 0.51 | 0.832 | 6.22 ± 0.66 | 5.14 ± 0.46 | 0.189 | 6.63 ± 0.54 | 5.85 ± 0.34 | 0.225 |
| Processed Meat | 1.4 ± 0.3 | 1.06 ± 0.16 | 0.335 | 1.1 ± 0.22 | 0.75 ± 0.15 | 0.202 | 1.21 ± 0.17 | 0.89 ± 0.11 | 0.133 |
| Poultry | 3.61 ± 0.86 | 5.2 ± 0.48 | 0.113 | 2.95 ± 0.53 | 4.22 ± 0.37 | 0.053 | 3.22 ± 0.46 | 4.64 ± 0.29 | **0.011** |
| Fish | 0.9 ± 0.49 | 1.7 ± 0.28 | 0.167 | 0.86 ± 0.25 | 1 ± 0.17 | 0.651 | 0.91 ± 0.24 | 1.3 ± 0.15 | 0.192 |
| Eggs | 1.2 ± 0.21 | 0.68 ± 0.11 | **0.033** | 1.29 ± 0.16 | 0.67 ± 0.11 | **0.002** | 1.27 ± 0.12 | 0.67 ± 0.08 | **<0.001** |
| Fruits, Total | | | | | | | | | |
| Fruits | 5.34 ± 0.42 | 2.49 ± 0.24 | **<0.001** | 7.16 ± 0.4 | 4.15 ± 0.27 | **<0.001** | 6.38 ± 0.29 | 3.46 ± 0.18 | **<0.001** |
| Fresh Fruit Juices | 0.34 ± 0.12 | 0.2 ± 0.06 | 0.301 | 0.59 ± 0.11 | 0.27 ± 0.07 | 0.019 | 0.49 ± 0.08 | 0.24 ± 0.05 | **0.012** |
| Sweets and Added Sugars | | | | | | | | | |
| Sweets | 6.16 ± 0.8 | 5.96 ± 0.44 | 0.830 | 8.56 ± 0.74 | 9.14 ± 0.52 | 0.526 | 7.56 ± 0.55 | 7.79 ± 0.35 | 0.728 |
| Added Sugars, Jams, Honey, Molasses | 1.38 ± 0.21 | 1.46 ± 0.11 | 0.737 | 1.7 ± 0.18 | 1.76 ± 0.12 | 0.779 | 1.58 ± 0.13 | 1.63 ± 0.08 | 0.738 |
| Sugar Sweetened Beverages | 5.52 ± 0.49 | 5.22 ± 0.27 | 0.599 | 4.45 ± 0.34 | 4.1 ± 0.24 | 0.408 | 4.9 ± 0.28 | 4.57 ± 0.18 | 0.328 |
| Hot Beverages (Coffee, Tea) | 0.15 ± 0.1 | 0.53 ± 0.06 | **0.002** | 0.63 ± 0.15 | 0.95 ± 0.1 | 0.083 | 0.44 ± 0.1 | 0.77 ± 0.06 | **0.005** |
| Alcoholic Beverages | 0.98 ± 0.35 | 1.05 ± 0.19 | 0.873 | 0.31 ± 0.1 | 0.25 ± 0.07 | 0.599 | 0.58 ± 0.14 | 0.59 ± 0.09 | 0.982 |
| Added Fats and Oils | 2.37 ± 0.43 | 3.59 ± 0.24 | **0.014** | 2.88 ± 0.36 | 3.72 ± 0.25 | 0.060 | 2.69 ± 0.27 | 3.67 ± 0.17 | **0.003** |
| Fast Food | 3.98 ± 0.93 | 4.13 ± 0.52 | 0.891 | 1.77 ± 0.51 | 2.21 ± 0.36 | 0.491 | 2.63 ± 0.47 | 3.04 ± 0.3 | 0.475 |
| Miscellaneous | 0.37 ± 0.29 | 1.12 ± 0.16 | **0.028** | 0.63 ± 0.21 | 1.06 ± 0.14 | 0.099 | 0.55 ± 0.17 | 1.08 ± 0.1 | **0.009** |

**Table 2.** *Cont.*

| | Males (n = 1572) | | | Females (n = 2009) | | | Both Sexes (n = 3581) | | |
|---|---|---|---|---|---|---|---|---|---|
| | 1997 (n = 436) | 2008/2009 (n = 1136) | *p*-Value | 1997 (n = 627) | 2008/2009 (n = 1382) | *p*-Value | 1997 (n = 1063) | 2008/2009 (n = 2518) | *p*-Value |
| **40–59.9 years** | | | | | | | | | |
| Bread | 26.45 ± 1.19 | 21.7 ± 0.84 | **0.001** | 19.3 ± 0.96 | 17.52 ± 0.74 | 0.151 | 22.58 ± 0.74 | 19.37 ± 0.55 | **0.001** |
| Cereals and Cereal-Based Products | 8.92 ± 1.54 | 15.53 ± 1.08 | **0.001** | 11.56 ± 1.36 | 15.32 ± 1.06 | **0.034** | 10.6 ± 1.01 | 15.33 ± 0.75 | **<0.001** |
| Legumes | 3.08 ± 0.8 | 4 ± 0.56 | 0.350 | 3.01 ± 0.63 | 3.05 ± 0.49 | 0.962 | 3.03 ± 0.49 | 3.51 ± 0.36 | 0.437 |
| Starchy Vegetables | 3.96 ± 0.52 | 3.32 ± 0.37 | 0.329 | 3.95 ± 0.51 | 3.36 ± 0.4 | 0.375 | 3.96 ± 0.36 | 3.34 ± 0.27 | 0.171 |
| Vegetables | 8.45 ± 0.79 | 7.44 ± 0.56 | 0.304 | 11.22 ± 0.99 | 11.51 ± 0.77 | 0.821 | 9.93 ± 0.64 | 9.64 ± 0.48 | 0.714 |
| Chips and Salty Crackers | 0.04 ± 0.13 | 0.33 ± 0.09 | 0.084 | 0.18 ± 0.13 | 0.3 ± 0.1 | 0.497 | 0.12 ± 0.09 | 0.31 ± 0.07 | 0.131 |
| Nuts and Seeds | 2.69 ± 0.67 | 2.6 ± 0.47 | 0.914 | 2.24 ± 0.61 | 2.56 ± 0.48 | 0.692 | 2.46 ± 0.44 | 2.56 ± 0.33 | 0.862 |
| Dairy Products | | | | | | | | | |
| Milk | 0.85 ± 0.18 | 0.48 ± 0.12 | 0.103 | 1.94 ± 0.28 | 1.37 ± 0.22 | 0.129 | 1.4 ± 0.17 | 0.99 ± 0.13 | 0.062 |
| Milk Derivatives | 7.54 ± 0.7 | 6.77 ± 0.49 | 0.376 | 9.19 ± 0.67 | 7.35 ± 0.52 | **0.036** | 8.35 ± 0.48 | 7.16 ± 0.36 | **0.049** |
| Sweetened Milk | 0.05 ± 0.06 | 0.09 ± 0.04 | 0.606 | 0.27 ± 0.2 | 0.35 ± 0.15 | 0.764 | 0.14 ± 0.11 | 0.24 ± 0.08 | 0.468 |
| Meat, Processed Meat, Poultry, Fish, Eggs | | | | | | | | | |
| Red Meat | 8.22 ± 1.02 | 7.14 ± 0.72 | 0.395 | 6.8 ± 0.88 | 7.37 ± 0.69 | 0.614 | 7.41 ± 0.66 | 7.26 ± 0.49 | 0.852 |
| Processed Meat | 1.1 ± 0.3 | 0.79 ± 0.21 | 0.412 | 0.47 ± 0.16 | 0.5 ± 0.12 | 0.891 | 0.78 ± 0.16 | 0.61 ± 0.12 | 0.421 |
| Poultry | 1.84 ± 0.65 | 3.54 ± 0.46 | **0.036** | 2.46 ± 0.53 | 2.86 ± 0.41 | 0.562 | 2.17 ± 0.41 | 3.2 ± 0.3 | **0.048** |
| Fish | 1.66 ± 0.4 | 1.14 ± 0.28 | 0.298 | 0.61 ± 0.28 | 0.79 ± 0.22 | 0.623 | 1.06 ± 0.23 | 0.96 ± 0.17 | 0.741 |
| Eggs | 1.75 ± 0.29 | 1.14 ± 0.2 | 0.092 | 1.68 ± 0.26 | 0.55 ± 0.2 | **0.001** | 1.72 ± 0.19 | 0.82 ± 0.14 | **<0.001** |
| Fruits, Total | | | | | | | | | |
| Fruits | 7.38 ± 0.49 | 4.21 ± 0.35 | **<0.001** | 8.55 ± 0.59 | 6.23 ± 0.46 | **0.003** | 7.96 ± 0.39 | 5.32 ± 0.29 | **<0.001** |
| Fresh Fruit Juices | 0.55 ± 0.14 | 0.23 ± 0.1 | 0.080 | 0.25 ± 0.07 | 0.16 ± 0.05 | 0.301 | 0.41 ± 0.07 | 0.18 ± 0.05 | **0.014** |
| Sweets and Added Sugars | | | | | | | | | |
| Sweets | 3.5 ± 0.72 | 4.12 ± 0.51 | 0.493 | 5.93 ± 0.83 | 5.89 ± 0.65 | 0.972 | 4.73 ± 0.55 | 5.14 ± 0.42 | 0.560 |
| Added Sugars, Jams, Honey, Molasses | 1.09 ± 0.23 | 1.66 ± 0.16 | **0.042** | 1.11 ± 0.18 | 1.81 ± 0.14 | **0.004** | 1.11 ± 0.14 | 1.74 ± 0.1 | **0.001** |
| Sugar Sweetened Beverages | 2.99 ± 0.45 | 2.79 ± 0.32 | 0.729 | 3.24 ± 0.51 | 2.85 ± 0.4 | 0.550 | 3.1 ± 0.34 | 2.84 ± 0.25 | 0.554 |
| Hot Beverages (Coffee, Tea) | 0.27 ± 0.09 | 0.47 ± 0.06 | 0.084 | 0.35 ± 0.07 | 0.47 ± 0.05 | 0.185 | 0.34 ± 0.05 | 0.46 ± 0.04 | 0.087 |
| Alcoholic Beverages | 2.04 ± 0.43 | 1.58 ± 0.3 | 0.395 | 0.21 ± 0.07 | 0.13 ± 0.06 | 0.435 | 1.04 ± 0.19 | 0.79 ± 0.14 | 0.308 |
| Added Fats and Oils | 3.6 ± 0.55 | 5.77 ± 0.39 | **0.002** | 4.19 ± 0.54 | 5.15 ± 0.42 | 0.177 | 3.93 ± 0.38 | 5.45 ± 0.29 | **0.002** |
| Fast Food | 1.28 ± 0.59 | 1.79 ± 0.41 | 0.485 | 0.64 ± 0.4 | 1.04 ± 0.31 | 0.437 | 0.91 ± 0.33 | 1.4 ± 0.25 | 0.252 |
| Miscellaneous | 0.59 ± 0.32 | 1.22 ± 0.23 | 0.114 | 0.55 ± 0.31 | 1.39 ± 0.24 | **0.037** | 0.62 ± 0.22 | 1.28 ± 0.16 | **0.019** |

**Table 2.** *Cont.*

| | Males (n = 1572) | | | Females (n = 2009) | | | Both Sexes (n = 3581) | | |
|---|---|---|---|---|---|---|---|---|---|
| | 1997 (n = 436) | 2008/2009 (n = 1136) | *p*-Value | 1997 (n = 627) | 2008/2009 (n = 1382) | *p*-Value | 1997 (n = 1063) | 2008/2009 (n = 2518) | *p*-Value |
| **≥60 years** | | | | | | | | | |
| Bread | 28.74 ± 1.57 | 22.51 ± 1.13 | **0.002** | 23.37 ± 1.57 | 19.85 ± 1.15 | 0.076 | 26.45 ± 1.11 | 21.14 ± 0.81 | **<0.001** |
| Cereals and Cereal-Based Products | 9.12 ± 1.8 | 12.74 ± 1.29 | 0.108 | 11.73 ± 2.03 | 16.56 ± 1.49 | 0.058 | 10.14 ± 1.34 | 14.63 ± 0.97 | **0.007** |
| Legumes | 3.88 ± 0.98 | 4.23 ± 0.7 | 0.778 | 2.43 ± 0.97 | 2.91 ± 0.71 | 0.691 | 3.23 ± 0.68 | 3.6 ± 0.5 | 0.672 |
| Starchy Vegetables | 3.13 ± 0.57 | 2.64 ± 0.41 | 0.499 | 3.33 ± 0.81 | 2.92 ± 0.59 | 0.688 | 3.22 ± 0.48 | 2.77 ± 0.35 | 0.464 |
| Vegetables | 9.16 ± 1.07 | 8.66 ± 0.77 | 0.710 | 10.65 ± 1.43 | 12.33 ± 1.05 | 0.348 | 9.79 ± 0.87 | 10.4 ± 0.63 | 0.572 |
| Chips and Salty Crackers | 0.1 ± 0.24 | 0.25 ± 0.17 | 0.627 | 0.01 ± 0.1 | 0.13 ± 0.07 | 0.252 | 0.04 ± 0.13 | 0.2 ± 0.1 | 0.375 |
| Nuts and Seeds | 1.7 ± 0.59 | 1.52 ± 0.43 | 0.814 | 0.83 ± 0.35 | 0.51 ± 0.26 | 0.473 | 1.33 ± 0.35 | 1.03 ± 0.25 | 0.493 |
| Dairy Products | | | | | | | | | |
| Milk | 1.92 ± 0.38 | 1.03 ± 0.27 | 0.065 | 2.6 ± 0.55 | 1.83 ± 0.4 | 0.265 | 2.2 ± 0.32 | 1.42 ± 0.23 | 0.059 |
| Milk Derivatives | 8.72 ± 0.91 | 7.89 ± 0.65 | 0.464 | 8.88 ± 1.01 | 9.96 ± 0.74 | 0.397 | 8.89 ± 0.67 | 8.8 ± 0.49 | 0.922 |
| Sweetened Milk | 0 ± 0.17 | 0.23 ± 0.12 | 0.265 | 0.02 ± 0.28 | 0.35 ± 0.21 | 0.365 | 0.01 ± 0.16 | 0.29 ± 0.11 | 0.173 |
| Meat, Processed Meat, Poultry, Fish, Eggs | | | | | | | | | |
| Red Meat | 9.74 ± 1.36 | 8.48 ± 0.98 | 0.459 | 6.88 ± 1.32 | 6.44 ± 0.97 | 0.789 | 8.38 ± 0.94 | 7.54 ± 0.69 | 0.477 |
| Processed Meat | 0.06 ± 0.28 | 0.42 ± 0.2 | 0.325 | 0.15 ± 0.32 | 0.62 ± 0.23 | 0.250 | 0.1 ± 0.21 | 0.51 ± 0.15 | 0.116 |
| Poultry | 2.06 ± 0.66 | 2.86 ± 0.47 | 0.331 | 2.03 ± 0.84 | 3.48 ± 0.62 | 0.169 | 2.12 ± 0.53 | 3.11 ± 0.38 | 0.136 |
| Fish | 0.38 ± 0.23 | 0.45 ± 0.16 | 0.801 | 1.53 ± 0.43 | 0.44 ± 0.31 | **0.043** | 0.94 ± 0.23 | 0.44 ± 0.17 | 0.089 |
| Eggs | 1.48 ± 0.43 | 1.08 ± 0.31 | 0.454 | 1.01 ± 0.26 | 0.47 ± 0.19 | 0.113 | 1.24 ± 0.26 | 0.81 ± 0.19 | 0.186 |
| Fruits, Total | | | | | | | | | |
| Fruits | 7.61 ± 0.81 | 6.52 ± 0.58 | 0.287 | 11.4 ± 0.97 | 8.23 ± 0.71 | **0.010** | 9.33 ± 0.62 | 7.35 ± 0.45 | **0.011** |
| Fresh Fruit Juices | 0.07 ± 0.11 | 0.22 ± 0.07 | 0.268 | 0.41 ± 0.17 | 0.29 ± 0.12 | 0.600 | 0.22 ± 0.09 | 0.26 ± 0.07 | 0.714 |
| Sweets and Added Sugars | | | | | | | | | |
| Sweets | 2.06 ± 0.94 | 4.65 ± 0.68 | **0.029** | 3.62 ± 0.95 | 3.22 ± 0.69 | 0.737 | 2.73 ± 0.67 | 4.02 ± 0.48 | 0.125 |
| Added Sugars, Jams, Honey, Molasses | 2.07 ± 0.33 | 1.62 ± 0.23 | 0.277 | 1.41 ± 0.23 | 1.17 ± 0.16 | 0.407 | 1.69 ± 0.2 | 1.45 ± 0.15 | 0.349 |
| Sugar Sweetened Beverages | 1.39 ± 0.37 | 1.32 ± 0.27 | 0.881 | 1.82 ± 0.5 | 1.79 ± 0.37 | 0.964 | 1.56 ± 0.3 | 1.56 ± 0.22 | 0.997 |
| Hot Beverages (Coffee, Tea) | 0.16 ± 0.02 | 0.2 ± 0.01 | 0.181 | 0.11 ± 0.04 | 0.26 ± 0.03 | **0.006** | 0.14 ± 0.02 | 0.23 ± 0.01 | **0.002** |
| Alcoholic Beverages | 1.81 ± 0.5 | 1.75 ± 0.36 | 0.919 | 0.1 ± 0.19 | 0.4 ± 0.14 | 0.210 | 1.05 ± 0.27 | 1.1 ± 0.2 | 0.898 |
| Added Fats and Oils | 4.27 ± 0.84 | 7.08 ± 0.6 | **0.008** | 4.01 ± 0.76 | 4.56 ± 0.56 | 0.568 | 4.19 ± 0.57 | 5.88 ± 0.41 | **0.018** |
| Fast Food | 0.01 ± 0.4 | 0.58 ± 0.29 | 0.260 | 0.79 ± 0.4 | 0.18 ± 0.3 | 0.228 | 0.37 ± 0.28 | 0.4 ± 0.21 | 0.936 |
| Miscellaneous | 0.27 ± 0.26 | 0.96 ± 0.18 | **0.032** | 0.8 ± 0.4 | 0.98 ± 0.29 | 0.728 | 0.54 ± 0.23 | 0.96 ± 0.17 | 0.144 |

**Table 2.** *Cont.*

| | Males (n = 1572) | | | Females (n = 2009) | | | Both Sexes (n = 3581) | | |
|---|---|---|---|---|---|---|---|---|---|
| | 1997 (n = 436) | 2008/2009 (n = 1136) | *p*-Value | 1997 (n = 627) | 2008/2009 (n = 1382) | *p*-Value | 1997 (n = 1063) | 2008/2009 (n = 2518) | *p*-Value |
| **Total Population (≥20 years)** | | | | | | | | | |
| Bread | 25.28 ± 0.74 | 19.82 ± 0.47 | **<0.001** | 18.88 ± 0.56 | 16.38 ± 0.41 | **<0.001** | 21.81 ± 0.45 | 17.84 ± 0.31 | **<0.001** |
| Cereals and Cereal-Based Products | 10.16 ± 0.94 | 15.72 ± 0.6 | **<0.001** | 12.72 ± 0.8 | 15.41 ± 0.58 | **0.007** | 11.45 ± 0.61 | 15.66 ± 0.42 | **<0.001** |
| Legumes | 3.4 ± 0.48 | 4.05 ± 0.31 | 0.248 | 2.39 ± 0.38 | 3.07 ± 0.28 | 0.150 | 2.87 ± 0.3 | 3.5 ± 0.21 | 0.080 |
| Starchy Vegetables | 4.12 ± 0.32 | 3.45 ± 0.21 | 0.080 | 4.25 ± 0.32 | 4.01 ± 0.23 | 0.532 | 4.2 ± 0.23 | 3.74 ± 0.16 | 0.099 |
| Vegetables | 7.64 ± 0.46 | 6.91 ± 0.29 | 0.182 | 10.11 ± 0.54 | 10.51 ± 0.39 | 0.554 | 8.95 ± 0.36 | 8.89 ± 0.25 | 0.892 |
| Chips and Salty Crackers | 0.12 ± 0.14 | 0.71 ± 0.09 | **<0.001** | 0.4 ± 0.14 | 0.91 ± 0.1 | **0.003** | 0.31 ± 0.1 | 0.81 ± 0.07 | **<0.001** |
| Nuts and Seeds | 2.9 ± 0.37 | 2.06 ± 0.24 | 0.060 | 2.24 ± 0.32 | 2.33 ± 0.23 | 0.808 | 2.53 ± 0.24 | 2.19 ± 0.17 | 0.251 |
| **Dairy Products** | | | | | | | | | |
| Milk | 1.15 ± 0.14 | 0.75 ± 0.09 | **0.017** | 1.87 ± 0.17 | 1.36 ± 0.12 | **0.018** | 1.53 ± 0.11 | 1.09 ± 0.08 | **0.001** |
| Milk Derivatives | 7.09 ± 0.43 | 6.77 ± 0.27 | 0.536 | 7.8 ± 0.39 | 7.99 ± 0.28 | 0.688 | 7.5 ± 0.29 | 7.42 ± 0.2 | 0.822 |
| Sweetened Milk | 0.02 ± 0.07 | 0.19 ± 0.04 | **0.040** | 0.15 ± 0.1 | 0.32 ± 0.07 | 0.189 | 0.09 ± 0.06 | 0.26 ± 0.04 | **0.022** |
| **Meat, Processed Meat, Poultry, Fish, Eggs** | | | | | | | | | |
| Red Meat | 7.93 ± 0.61 | 7.32 ± 0.39 | 0.400 | 6.51 ± 0.49 | 6.01 ± 0.36 | 0.416 | 7.16 ± 0.38 | 6.59 ± 0.26 | 0.219 |
| Processed Meat | 1.05 ± 0.18 | 0.84 ± 0.11 | 0.319 | 0.76 ± 0.14 | 0.65 ± 0.1 | 0.508 | 0.88 ± 0.11 | 0.74 ± 0.08 | 0.270 |
| Poultry | 2.73 ± 0.46 | 4.21 ± 0.3 | **0.008** | 2.64 ± 0.35 | 3.71 ± 0.26 | **0.014** | 2.68 ± 0.28 | 3.95 ± 0.19 | **<0.001** |
| Fish | 1.12 ± 0.26 | 1.28 ± 0.17 | 0.609 | 0.87 ± 0.18 | 0.87 ± 0.13 | 0.967 | 0.98 ± 0.15 | 1.06 ± 0.1 | 0.650 |
| Eggs | 1.44 ± 0.17 | 0.92 ± 0.11 | **0.008** | 1.38 ± 0.13 | 0.61 ± 0.09 | **<0.001** | 1.44 ± 0.1 | 0.74 ± 0.07 | **<0.001** |
| **Fruits, Total** | | | | | | | | | |
| Fruits | 6.42 ± 0.31 | 3.85 ± 0.2 | **<0.001** | 8.17 ± 0.32 | 5.41 ± 0.23 | **<0.001** | 7.36 ± 0.22 | 4.72 ± 0.15 | **<0.001** |
| Fresh Fruit Juices | 0.38 ± 0.08 | 0.21 ± 0.05 | 0.078 | 0.45 ± 0.07 | 0.24 ± 0.05 | **0.011** | 0.42 ± 0.05 | 0.23 ± 0.04 | **0.002** |
| **Sweets and Added Sugars** | | | | | | | | | |
| Sweets | 4.33 ± 0.48 | 5.11 ± 0.3 | 0.170 | 7.08 ± 0.5 | 7.25 ± 0.36 | 0.788 | 5.84 ± 0.35 | 6.3 ± 0.24 | 0.279 |
| Added Sugars, Jams, Honey, Molasses | 1.36 ± 0.14 | 1.59 ± 0.09 | 0.187 | 1.47 ± 0.12 | 1.69 ± 0.09 | 0.139 | 1.44 ± 0.09 | 1.64 ± 0.06 | 0.079 |
| Sugar Sweetened Beverages | 3.84 ± 0.28 | 3.67 ± 0.18 | 0.606 | 3.71 ± 0.26 | 3.37 ± 0.19 | 0.282 | 3.75 ± 0.19 | 3.52 ± 0.13 | 0.314 |
| Hot Beverages (Coffee, Tea) | 0.22 ± 0.06 | 0.45 ± 0.04 | **0.001** | 0.47 ± 0.08 | 0.71 ± 0.06 | **0.019** | 0.37 ± 0.05 | 0.59 ± 0.04 | **0.001** |
| Alcoholic Beverages | 1.53 ± 0.24 | 1.36 ± 0.15 | 0.555 | 0.25 ± 0.07 | 0.24 ± 0.05 | 0.879 | 0.83 ± 0.11 | 0.74 ± 0.08 | 0.502 |
| Added Fats and Oils | 3.14 ± 0.33 | 4.97 ± 0.21 | **<0.001** | 3.49 ± 0.28 | 4.29 ± 0.21 | **0.025** | 3.38 ± 0.21 | 4.59 ± 0.15 | **<0.001** |
| Fast Food | 2.23 ± 0.46 | 2.69 ± 0.3 | 0.410 | 1.28 ± 0.31 | 1.55 ± 0.22 | 0.478 | 1.67 ± 0.26 | 2.09 ± 0.18 | 0.193 |
| Miscellaneous | 0.42 ± 0.18 | 1.13 ± 0.11 | **0.001** | 0.66 ± 0.16 | 1.14 ± 0.12 | **0.019** | 0.58 ± 0.12 | 1.12 ± 0.08 | **<0.001** |

Values in this table represent means ± standard error (SE). The analysis of covariance (ANCOVA) was used to estimate the adjusted means of dietary intake. Numbers in **bold** face are statistically significant (*p*-value ≤ 0.05). Abbreviations: EI: energy intake.

*3.3. Energy (kcal) and Macronutrient Intake (%EI)*

The adjusted means ± standard error (SE) of total energy (kcal) and macronutrient intakes (expressed as contribution to total energy intake, %EI) (by sex and age) and their comparison between 1997 and 2008/2009 are shown in Table 3. Additional data describing the absolute intake of macronutrients (expressed in (g)) are presented in Supplementary Table S2.

Overall, a significant increase in the daily energy intake from 1728 ± 24 kcal in 1997 to 1877 ± 15 kcal in 2008/2009 was observed. The percentage of contribution of carbohydrates and fats to total energy varied significantly among the survey years and across the different age groups. Compared to 1997, in 2008/2009, carbohydrate intake decreased while that of fat increased significantly. In 2008/2009, the percent contribution of carbohydrates to total energy was 48.97 ± 0.23% compared to 51.32 ± 0.36% in 1997. The percentage of total fat intake also increased from 34.63 ± 0.32% in 1997 to 36.97 ± 0.21% in 2008/2009. Among the types of fat studied, the percentage of oleic acid intake increased from 10.73 ± 0.21% to 11.98 ± 0.13%, so did linoleic acid intake (from 3.6 ± 0.11% to 4.86 ± 0.07%) and saturated fat (from 9.97 ± 0.15% to 10.79 ± 0.1% EI). There were no significant changes in the contribution of protein to total energy intake between the two survey years. The percentage of total sugar intake significantly decreased from 11.48 ± 0.26% in 1997 to 10.83 ± 0.17% in 2008/2009. The aforementioned differences in overall dietary intake were also shown among males and females, except for proteins and sugars. More specifically, in males, a decrease in percent contribution of proteins was noted in 2008/2009 compared to 1997 (14.83 ± 0.23% vs. 15.76 ± 0.37%, $p < 0.05$), while no such difference was observed among females. Furthermore, while the decrease in sugar intake was significant among females (11.07 ± 0.23% in 2008/2009 vs. 11.97 ± 0.34% in 1997, $p < 0.05$), it was not among males (Table 3).

*3.4. Micronutrient Density (per 1000 Kcal)*

The adjusted means of micronutrient density (expressed per 1000 kcal intake) (by sex and age) and their comparison between 1997 and 2008/2009 are shown in Table 4. Additional data describing the absolute intake of micronutrients (expressed in (g)) are presented in Supplementary Table S3.

In the overall study population, compared to 1997, in 2008/2009 significant decreases in the micronutrient density of vitamin A and vitamin C were observed (vitamin A: 565.37 ± 42.17 µg/1000 kcal vs. 455.34 ± 27.25 µg/1000 kcal; vitamin C: 52.76 ± 1.36 mg/1000 kcal vs. 46.08 ± 0.88 mg/1000 kcal, in 1997 and 2008/2009, respectively). The changes in vitamin C were observed among participants aged 20–39.9 years and 40–59.9 years, and among females. In addition, comparing 1997 and 2008/2009, an observed significant increase in the intake of zinc was found among females (4.29 ± 0.09 mg/1000 kcal vs. 4.54 ± 0.06 mg/1000 kcal, respectively) and among participants aged ≥60 years (4.26 ± 0.14 mg/1000 kcal vs. 4.61 ± 0.09 mg/1000 kcal, respectively). Moreover, although not observed in the overall study population, calcium intake significantly decreased among participants aged 40–59.9 years (346.52 ± 10.44 mg/1000 kcal in 1997 vs. 316 ± 7.28 mg/1000 kcal in 2008/2009). No significant differences in micronutrient intakes were observed among males between the two years (Table 4).

**Table 3.** Mean daily total energy (kcal) and macronutrient intake (%EI) (by sex and age) and their comparison between 1997 and 2008/2009, in the study population (*n* = 3581), aged ≥20 years, Lebanon.

| | Males (n = 1572) | | | Females (n = 2009) | | | Both Sexes (n = 3581) | | |
|---|---|---|---|---|---|---|---|---|---|
| | 1997 (n = 436) | 2008/2009 (n = 1136) | *p*-Value | 1997 (n = 627) | 2008/2009 (n = 1382) | p-Value | 1997 (n = 1063) | 2008/2009 (n = 2518) | *p*-Value |
| **20–39.9 years** | | | | | | | | | |
| Energy (kcal) | 2259.96 ± 72.59 | 2343.40 ± 39.11 | 0.315 | 1529.98 ± 39.75 | 1657.84 ± 25.87 | **0.007** | 1838.92 ± 37.05 | 1937.00 ± 22.41 | **0.024** |
| Carbohydrate (%EI) | 52.1 ± 0.82 | 48.95 ± 0.44 | **0.001** | 51.65 ± 0.65 | 48.71 ± 0.42 | **<0.001** | 51.81 ± 0.51 | 48.81 ± 0.3 | **<0.001** |
| Protein (%EI) | 16.08 ± 0.75 | 14.85 ± 0.4 | 0.156 | 14.31 ± 0.3 | 14.26 ± 0.19 | 0.884 | 14.97 ± 0.34 | 14.5 ± 0.2 | 0.243 |
| Fat (%EI) | 33.33 ± 0.74 | 36.21 ± 0.39 | **0.001** | 35.27 ± 0.59 | 38.16 ± 0.38 | **<0.001** | 34.51 ± 0.46 | 37.34 ± 0.27 | **<0.001** |
| Saturated Fat (%EI) | 9.52 ± 0.33 | 10.79 ± 0.18 | **0.001** | 10.16 ± 0.28 | 11.54 ± 0.18 | **<0.001** | 9.9 ± 0.21 | 11.22 ± 0.13 | **<0.001** |
| Oleic Acid (%EI) | 9.61 ± 0.46 | 11.13 ± 0.24 | **0.004** | 10.71 ± 0.38 | 11.68 ± 0.24 | **0.035** | 10.3 ± 0.29 | 11.45 ± 0.17 | **0.001** |
| Linolenic Acid (%EI) | 0.11 ± 0 | 0.14 ± 0 | **<0.001** | 0.12 ± 0 | 0.15 ± 0 | **<0.001** | 0.12 ± 0 | 0.15 ± 0 | **<0.001** |
| Linoleic Acid (%EI) | 3.42 ± 0.24 | 4.76 ± 0.13 | **<0.001** | 3.64 ± 0.21 | 5.12 ± 0.14 | **<0.001** | 3.55 ± 0.16 | 4.97 ± 0.09 | **<0.001** |
| Total Sugar (%EI) | 11.74 ± 0.63 | 10.99 ± 0.33 | 0.297 | 12.33 ± 0.45 | 11.04 ± 0.29 | **0.018** | 12.12 ± 0.36 | 11.02 ± 0.22 | **0.011** |
| Dietary fibers (g) | 19.53 ± 1.17 | 18.07 ± 0.63 | 0.278 | 14.03 ± 0.56 | 14.36 ± 0.36 | 0.624 | 16.37 ± 0.56 | 15.83 ± 0.34 | 0.421 |
| **40–59.9 years** | | | | | | | | | |
| Energy (kcal) | 2016.51 ± 66.96 | 2269.91 ± 46.01 | **0.002** | 1387.53 ± 45.48 | 1631.11 ± 31.79 | **<0.001** | 1653.81 ± 38.53 | 1917.31 ± 26.88 | **<0.001** |
| Carbohydrate (%EI) | 50.15 ± 0.88 | 47.79 ± 0.6 | **0.029** | 50.44 ± 0.84 | 49.88 ± 0.58 | 0.595 | 50.46 ± 0.6 | 48.88 ± 0.42 | **0.033** |
| Protein (%EI) | 15.9 ± 0.43 | 15.03 ± 0.29 | 0.103 | 14.93 ± 0.61 | 14.83 ± 0.42 | 0.895 | 15.39 ± 0.38 | 14.9 ± 0.26 | 0.302 |
| Fat (%EI) | 33.66 ± 0.76 | 37.02 ± 0.52 | **<0.001** | 36.74 ± 0.77 | 36.39 ± 0.54 | 0.718 | 35.22 ± 0.54 | 36.77 ± 0.38 | **0.020** |
| Saturated Fat (%EI) | 9.78 ± 0.49 | 10.23 ± 0.33 | 0.456 | 10.54 ± 0.33 | 10.23 ± 0.23 | 0.458 | 10.14 ± 0.28 | 10.26 ± 0.19 | 0.728 |
| Oleic Acid (%EI) | 10.69 ± 0.54 | 12.67 ± 0.37 | **0.003** | 11.79 ± 0.49 | 12.34 ± 0.34 | 0.363 | 11.24 ± 0.36 | 12.52 ± 0.25 | **0.004** |
| Linolenic Acid (%EI) | 0.12 ± 0 | 0.15 ± 0 | **0.045** | 0.15 ± 0.01 | 0.15 ± 0 | 0.849 | 0.14 ± 0 | 0.15 ± 0 | 0.112 |
| Linoleic Acid (%EI) | 3.42 ± 0.24 | 4.94 ± 0.16 | **<0.001** | 3.81 ± 0.28 | 4.95 ± 0.19 | **0.001** | 3.64 ± 0.18 | 4.95 ± 0.13 | **<0.001** |
| Total Sugar (%EI) | 10.89 ± 0.73 | 9.6 ± 0.5 | 0.145 | 11 ± 0.65 | 11.31 ± 0.45 | 0.704 | 10.92 ± 0.48 | 10.56 ± 0.33 | 0.541 |
| Dietary Fibers (g/d) | 18.43 ± 0.96 | 20.14 ± 0.66 | 0.147 | 14.46 ± 0.81 | 15.74 ± 0.56 | 0.207 | 16.14 ± 0.61 | 17.71 ± 0.42 | **0.037** |
| **≥60 years** | | | | | | | | | |
| Energy (kcal) | 1794.76 ± 70.98 | 1833.96 ± 48.35 | 0.651 | 1305.72 ± 66.96 | 1425.26 ± 43.94 | 0.139 | 1560.40 ± 48.79 | 1633.67 ± 32.70 | 0.215 |
| Carbohydrate (%EI) | 50.89 ± 1.21 | 48.59 ± 0.82 | 0.122 | 53.45 ± 1.24 | 50.34 ± 0.81 | **0.039** | 52.04 ± 0.86 | 49.48 ± 0.58 | **0.015** |
| Protein (%EI) | 14.85 ± 0.41 | 14.51 ± 0.28 | 0.514 | 14.49 ± 0.58 | 15.18 ± 0.38 | 0.326 | 14.71 ± 0.35 | 14.83 ± 0.23 | 0.774 |
| Fat (%EI) | 33.11 ± 1.06 | 37.02 ± 0.72 | **0.003** | 33.56 ± 1.08 | 35.57 ± 0.71 | 0.127 | 33.35 ± 0.75 | 36.29 ± 0.5 | **0.001** |
| Saturated Fat (%EI) | 9.39 ± 0.46 | 10.29 ± 0.31 | 0.113 | 9.83 ± 0.51 | 10.64 ± 0.34 | 0.196 | 9.62 ± 0.34 | 10.45 ± 0.23 | **0.046** |
| Oleic Acid (%EI) | 10.58 ± 0.69 | 13.03 ± 0.47 | **0.004** | 11.05 ± 0.69 | 12.15 ± 0.45 | 0.188 | 10.82 ± 0.49 | 12.59 ± 0.32 | **0.003** |
| Linolenic Acid (%EI) | 0.13 ± 0 | 0.16 ± 0 | **0.009** | 0.14 ± 0.01 | 0.16 ± 0 | 0.061 | 0.13 ± 0 | 0.16 ± 0 | **0.001** |
| Linoleic Acid (%EI) | 3.43 ± 0.37 | 4.73 ± 0.25 | **0.004** | 3.57 ± 0.32 | 3.97 ± 0.21 | **0.314** | 3.47 ± 0.24 | 4.36 ± 0.16 | **0.003** |
| Total Sugar (%EI) | 9.28 ± 0.86 | 10.67 ± 0.59 | 0.189 | 12.92 ± 0.87 | 10.67 ± 0.57 | 0.034 | 10.94 ± 0.61 | 10.7 ± 0.41 | 0.746 |
| Dietary Fibers (g/d) | 18.21 ± 1.18 | 17.56 ± 0.8 | 0.651 | 14.59 ± 1.1 | 16.43 ± 0.72 | 0.164 | 16.5 ± 0.8 | 17 ± 0.54 | 0.608 |

**Table 3.** *Cont.*

| | Males (n = 1572) | | | Females (n = 2009) | | | Both Sexes (n = 3581) | | |
|---|---|---|---|---|---|---|---|---|---|
| | 1997 (n = 436) | 2008/2009 (n = 1136) | *p*-Value | 1997 (n = 627) | 2008/2009 (n = 1382) | p-Value | 1997 (n = 1063) | 2008/2009 (n = 2518) | *p*-Value |
| **Total Population (≥20 years)** | | | | | | | | | |
| Energy (kcal) | 2080.54 ± 42.06 | 2216.95 ± 25.9 | **0.006** | 1452.76 ± 27.34 | 1612.22 ± 18.27 | **<0.001** | 1727.97 ± 23.78 | 1877.44 ± 15.38 | **<0.001** |
| Carbohydrate (%EI) | 51.11 ± 0.54 | 48.52 ± 0.33 | **<0.001** | 51.5 ± 0.48 | 49.32 ± 0.32 | **<0.001** | 51.32 ± 0.36 | 48.97 ± 0.23 | **<0.001** |
| Protein (%EI) | 15.76 ± 0.37 | 14.83 ± 0.23 | **0.033** | 14.55 ± 0.27 | 14.58 ± 0.18 | 0.925 | 15.05 ± 0.22 | 14.7 ± 0.14 | 0.175 |
| Fat (%EI) | 33.43 ± 0.47 | 36.61 ± 0.29 | **<0.001** | 35.52 ± 0.43 | 37.23 ± 0.29 | **0.001** | 34.63 ± 0.32 | 36.97 ± 0.21 | **<0.001** |
| Saturated Fat (%EI) | 9.64 ± 0.25 | 10.51 ± 0.15 | **0.003** | 10.24 ± 0.2 | 11 ± 0.13 | **0.001** | 9.97 ± 0.15 | 10.79 ± 0.1 | **<0.001** |
| Oleic Acid (%EI) | 10.19 ± 0.32 | 11.99 ± 0.19 | **<0.001** | 11.13 ± 0.28 | 11.97 ± 0.18 | **0.012** | 10.73 ± 0.21 | 11.98 ± 0.13 | **<0.001** |
| Linolenic Acid (%EI) | 0.12 ± 0.01 | 0.15 ± 0 | **<0.001** | 0.14 ± 0.01 | 0.16 ± 0 | **0.002** | 0.13 ± 0 | 0.16 ± 0 | **<0.001** |
| Linoleic Acid (%EI) | 3.44 ± 0.16 | 4.81 ± 0.1 | **<0.001** | 3.72 ± 0.15 | 4.89 ± 0.1 | **<0.001** | 3.6 ± 0.11 | 4.86 ± 0.07 | **<0.001** |
| Total Sugar (%EI) | 10.88 ± 0.42 | 10.5 ± 0.26 | 0.434 | 11.97 ± 0.34 | 11.07 ± 0.23 | **0.031** | 11.48 ± 0.26 | 10.83 ± 0.17 | **0.041** |
| Dietary Fibers (g/d) | 18.81 ± 0.66 | 18.61 ± 0.41 | 0.794 | 14.24 ± 0.43 | 15.12 ± 0.29 | 0.089 | 16.26 ± 0.37 | 16.64 ± 0.24 | 0.401 |

Values in this table represent means ± SE. The analysis of covariance (ANCOVA) was used to estimate adjusted means of dietary intake. Numbers in **bold** face are statistically significant (*p*-value ≤ 0.05). Abbreviations: EI: energy intake.

**Table 4.** Mean daily micronutrient density (by sex and age) and their comparison between 1997 and 2008/2009, in the study population ($n$ = 3581), aged ≥20 years, Lebanon.

| | Males (n = 1572) | | | Females (n = 2009) | | | Both Sexes (n = 3581) | | |
|---|---|---|---|---|---|---|---|---|---|
| | 1997 (n = 436) | 2008/2009 (n = 1136) | *p*-Value | 1997 (n = 627) | 2008/2009 (n = 1382) | *p*-Value | 1997 (n = 1063) | 2008/2009 (n = 2518) | *p*-Value |
| **20–39.9 years** | | | | | | | | | |
| Calcium (mg/1000 kcal intake) | 326.97 ± 17.33 | 313.32 ± 9.33 | 0.491 | 326.5 ± 10.39 | 328.65 ± 6.74 | 0.863 | 326.78 ± 9.2 | 321.91 ± 5.55 | 0.653 |
| Iron (mg/1000 kcal intake) | 5.02 ± 0.17 | 5.29 ± 0.09 | 0.182 | 5.35 ± 0.14 | 5.54 ± 0.09 | 0.261 | 5.2 ± 0.11 | 5.44 ± 0.06 | 0.069 |
| Zinc (mg/1000 kcal intake) | 4.57 ± 0.18 | 4.45 ± 0.1 | 0.563 | 4.18 ± 0.12 | 4.48 ± 0.08 | **0.049** | 4.32 ± 0.1 | 4.46 ± 0.06 | 0.238 |
| Vitamin A (µg/1000 kcal intake) | 345.68 ± 75.92 | 399.95 ± 40.9 | 0.532 | 551.13 ± 68.82 | 482.33 ± 44.66 | 0.404 | 469.96 ± 51.33 | 449.78 ± 30.98 | 0.738 |
| Vitamin C (mg/1000 kcal intake) | 40.24 ± 2.63 | 41.56 ± 1.42 | 0.660 | 61.81 ± 3.06 | 49.1 ± 1.98 | **0.001** | 53.64 ± 2.14 | 46.04 ± 1.29 | **0.003** |
| Vitamin B12 (µg/1000 kcal intake) | 1.75 ± 0.56 | 2.65 ± 0.3 | 0.169 | 2.11 ± 0.46 | 2.3 ± 0.3 | 0.731 | 1.98 ± 0.35 | 2.45 ± 0.21 | 0.264 |
| **40–59.9 years** | | | | | | | | | |
| Calcium (mg/1000 kcal intake) | 315.2 ± 13.84 | 292.4 ± 9.51 | 0.176 | 371.87 ± 15.39 | 334.07 ± 10.76 | **0.048** | 346.52 ± 10.44 | 316 ± 7.28 | **0.017** |
| Iron (mg/1000 kcal intake) | 5.35 ± 0.2 | 5.42 ± 0.14 | 0.791 | 5.57 ± 0.19 | 5.73 ± 0.13 | 0.530 | 5.48 ± 0.14 | 5.59 ± 0.09 | 0.523 |
| Zinc (mg/1000 kcal intake) | 4.7 ± 0.19 | 4.51 ± 0.13 | 0.431 | 4.5 ± 0.16 | 4.54 ± 0.11 | 0.848 | 4.57 ± 0.12 | 4.53 ± 0.08 | 0.785 |
| Vitamin A (µg/1000 kcal intake) | 652.09 ± 138.37 | 423.22 ± 95.08 | 0.175 | 668.08 ± 123.21 | 516.05 ± 86.14 | 0.320 | 654.87 ± 90.92 | 477.88 ± 63.43 | 0.112 |
| Vitamin C (mg/1000 kcal intake) | 46.65 ± 2.69 | 41.41 ± 1.85 | 0.110 | 60.24 ± 3.22 | 46.68 ± 2.25 | **0.001** | 54.43 ± 2.13 | 44.31 ± 1.48 | **<0.001** |
| Vitamin B12 (µg/1000 kcal intake) | 2.58 ± 0.74 | 2.73 ± 0.5 | 0.869 | 2.04 ± 0.64 | 2.73 ± 0.45 | 0.389 | 2.22 ± 0.48 | 2.75 ± 0.33 | 0.368 |
| **≥60 years** | | | | | | | | | |
| Calcium (mg/1000 kcal intake) | 358.15 ± 17.28 | 324.34 ± 11.77 | 0.109 | 377.72 ± 21.26 | 384.98 ± 13.94 | 0.777 | 368.01 ± 13.59 | 353.97 ± 9.1 | 0.393 |
| Iron (mg/1000 kcal intake) | 5.4 ± 0.27 | 5.43 ± 0.19 | 0.929 | 5.57 ± 0.52 | 6.03 ± 0.34 | 0.463 | 5.44 ± 0.29 | 5.74 ± 0.19 | 0.388 |
| Zinc (mg/1000 kcal intake) | 4.35 ± 0.2 | 4.46 ± 0.13 | 0.663 | 4.14 ± 0.2 | 4.78 ± 0.13 | **0.012** | 4.26 ± 0.14 | 4.61 ± 0.09 | **0.043** |
| Vitamin A (µg/1000 kcal intake) | 702.73 ± 140.39 | 439.04 ± 95.63 | 0.124 | 528.86 ± 80.12 | 429.95 ± 52.57 | 0.306 | 620.36 ± 81.89 | 434.54 ± 54.88 | 0.061 |
| Vitamin C (mg/1000 kcal intake) | 42.97 ± 3.75 | 43.48 ± 2.55 | 0.911 | 53.56 ± 4.44 | 54.83 ± 2.91 | 0.813 | 47.75 ± 2.88 | 49.18 ± 1.93 | 0.681 |
| Vitamin B12 (µg/1000 kcal intake) | 4.33 ± 1.2 | 3.03 ± 0.81 | 0.375 | 3.03 ± 0.68 | 1.84 ± 0.44 | 0.148 | 3.72 ± 0.7 | 2.44 ± 0.46 | 0.132 |
| **Total population (≥20 years)** | | | | | | | | | |
| Calcium (mg/1000 kcal intake) | 329.76 ± 9.64 | 309 ± 5.94 | 0.068 | 348.12 ± 8.07 | 339.59 ± 5.38 | 0.383 | 339.47 ± 6.17 | 326.27 ± 3.98 | 0.073 |
| Iron (mg/1000 kcal intake) | 5.22 ± 0.12 | 5.36 ± 0.07 | 0.323 | 5.47 ± 0.13 | 5.68 ± 0.08 | 0.172 | 5.34 ± 0.09 | 5.55 ± 0.06 | 0.056 |
| Zinc (mg/1000 kcal intake) | 4.58 ± 0.12 | 4.47 ± 0.07 | 0.415 | 4.29 ± 0.09 | 4.54 ± 0.06 | **0.022** | 4.41 ± 0.07 | 4.51 ± 0.05 | 0.198 |
| Vitamin A (µg/1000 kcal intake) | 531.85 ± 65.81 | 419.23 ± 40.52 | 0.147 | 590.62 ± 55.42 | 484.14 ± 36.97 | 0.113 | 565.37 ± 42.17 | 455.34 ± 27.25 | **0.029** |
| Vitamin C (mg/1000 kcal intake) | 43.26 ± 1.68 | 41.9 ± 1.04 | 0.492 | 59.91 ± 2.02 | 49.28 ± 1.35 | **<0.001** | 52.76 ± 1.36 | 46.08 ± 0.88 | **<0.001** |
| Vitamin B12 (µg/1000 kcal intake) | 2.59 ± 0.44 | 2.78 ± 0.27 | 0.716 | 2.23 ± 0.34 | 2.37 ± 0.23 | 0.749 | 2.39 ± 0.27 | 2.55 ± 0.17 | 0.630 |

Values in this table represent means ± SE. The analysis of covariance (ANCOVA) was used to estimate adjusted means of dietary intake. Numbers in **bold** face are statistically significant (*p*-value ≤ 0.05).

## 4. Discussion

To our knowledge, this study is the first to characterize the nutrition transition in the EMR, based on individual national nutrition surveys. The results of this study have documented significant changes in food consumption and nutrient intakes among Lebanese adults, between 1997 and 2008/2009. Overall, a downward trend was observed in the daily consumption, expressed as % energy intake (%EI), of bread, fruits, fresh fruit juices, milk and eggs, whereas the consumption of added fats and oils, poultry, cereals and cereal-based products, chips and crackers, sweetened milk and hot beverages followed an increasing trend over the 12-year study period. The results have also documented a significant increase in dietary energy (kcal/day) and fat intake (%EI), coupled with decreases in carbohydrate intake (%EI) and dietary density of vitamin A and vitamin C (per 1000 kcal). These changes in food consumption and nutrient intakes were observed in both genders and across age groups, with some disparities being noted between groups.

In many aspects, the observed shifts in the food groups' intakes are consistent with the hallmarks of the nutrition transition [22] and which are described as increases in the consumption of fat and animal-based products, contrasted with a fall in the intake of total cereal, fruit and vegetables [22]. The observed decrease in fruit juice intake in Lebanon is considered as a favorable dietary change, but the decline in the consumption of whole fruits (from 210 g/day in 1997 to 112 g/day in 2008/2009) is a point of concern. Fruits are rich sources of dietary fiber, antioxidants and phytochemicals and have been consistently associated with improved cardiometabolic health [1] and decreased NCD-related morbidity and mortality [1]. The recent Global Burden of Disease Study (GBD) [23] showed that the optimal intake of fruit in relation to all-cause mortality falls within the range of 200–300 g/day. The observed decrease in the intake of fruit over time highlights a priority area for public health intervention, given that low intake of fruits was reported as one of the dietary factors that are responsible for the highest number of CMD in Lebanon [1]. It is worth noting that, over the same period of time, the population's intake of vegetables decreased from 207 to 191 g/day, and this decrease was borderline significant ($p < 0.059$). The level of vegetable intake in 2008/2009 (191 g/day) is almost half the optimal range of intake recommended by the GBD (290–430 g/day), thus underscoring an additional priority area for intervention in Lebanon.

The study findings showed that intake of breads decreased over time in Lebanon, but the consumption of cereals and cereal-based products, which include refined-grain food items such as white rice, pizzas, pies and breakfast cereals, has increased over time, from 11.45% to 15.66% EI. These findings highlight carbohydrate quality as a potential point of concern in the population's diet, given the impact of refined grains on the glycemic response, insulin excursions and NCD risk [24]. Low intake of whole grains was in fact found to be among the leading risk factors for CMD mortality in Lebanon [1]. The results of our study have also documented a decrease in milk consumption, while the consumption of milk derivatives has remained quite stable over time. Together, the consumption of milk and its derivatives was estimated at 111.6 g/day in 1997, decreasing to 84.6 g/day in 2008/2009, which is short of the recommended optimal range of intake (350–520 g/day) [23].

Nutrition transition is usually characterized by increases in animal-based products. In our study, the consumption of poultry increased significantly over time (2.68% to 3.95% EI), but the intake of red meat and processed meat remained quite stable. However, despite this stability, the level of meat intake remained too high (42.26 g/day), being almost double the optimal range of intake defined by the GBD study (18–27 g/day) [23]. The same can be noted for processed meat consumption (4.49 g/day vs. an optimal range of 0–4 g/day) [23]. In addition, the consumption of sugar sweetened beverages (SSBs) did not increase over time, but its consumption remained very high (165.76 g/day) in comparison with recommended intake levels (0–5 g/day) [23]. High intakes of meat, processed meat and SSBs may be associated with an increased risk for various NCDs, including obesity, cardiovascular diseases and certain types of cancer. In a country where NCD mortality is estimated to represent 84% of total deaths [25], there is a need for population-based interventions and nutrition policies aimed at promoting healthier and more balanced diets.

The observed shifts in the consumption of various food groups were translated into changes in energy and nutrient intakes. There were significant increases in average daily energy intake, which may have implications on overweight and obesity trends in the population. A recent study based on the same national surveys has in fact documented an increase in the prevalence of adult obesity in Lebanon, from 17.4% to 28.2% between 1997 and 2008/2009 [12]. The contribution of the various macronutrients to dietary energy intake has also changed during the study period. Energy intakes from carbohydrates were found to decrease, while energy derived from fat increased over time. This pattern of change in macronutrients intake is typical of the nutrition transition and the dietary shifts that characterize it [22]. The surge in fat intake may be partially linked to the observed increases in the consumption of added fats and oils as well as increased intake of high fat cereal-based products such as pizzas and pies, in addition to chips and crackers. More importantly, the proportion of subjects exceeding the upper limit for fat (30% EI) [26] increased significantly from 63% to 70% during the study period, and the proportion of those exceeding the saturated fat upper limit (10% EI) [26] increased significantly from 38% to 51% (data not shown). In our study, the decline in total sugar intake reflect the decrease in the consumption of fruit and fresh fruit juices. This decrease may also explain the reduction in the dietary density of vitamin A and vitamin C (per 1000 kcal intake).

Given the lack of other national food consumption data in Lebanon, we compared our study findings to those derived from FBSs covering the same period of time (1997 vs. 2009) [27]. The results concerning the increase in total fat intake at the expense of carbohydrates was in agreement with FBS data, whereby the contribution of fat to dietary energy supply (DES) increased from 28.7% in 1997 to 31.7% in 2009, while that of carbohydrates decreased from 60% to 57% (data not shown). FBS data also documented an increase in vegetable oil intake in Lebanon (from 10.5% to 14% DES), which is in line with the study findings related to the increase in the intake of added fats and oils. The observed decreases in the consumption of fruits, fresh fruit juices, milk and eggs are also consistent with data stemming from FBSs. For instance, FBS data illustrated a decrease in the per capita supply of fruits, from 564 g/person/day in 1997 to 278 g/person/day in 2009, which represents a decrease from 7.38% to 4.23% DES (data not shown). In our study, no or very minor changes were found regarding the consumption of nuts, pulses, meat/meat products and fish/seafood. Similar trends were observed based on the FBS data. In contrast, FBS data show a sharp decrease in the vegetable supply (932 g/day in 1997 to 413.6 g/day in 2009) and an increase in the contribution of sugar and sweeteners to DES (from 11% to 13% DES between 1997 and 2009), which could not be observed in our study.

The shift in food consumption and nutrient intakes in Lebanon share similarities with that reported from other Mediterranean countries undergoing the nutrition transition [28,29]. For instance, and in agreement with our study findings, the dietary trend in Spain was towards a significant increase in daily fat intake, while an increase in the consumption of meat was also observed [28]. Based on household budget and consumption surveys, the consumption of fruits was found to decrease over time in Morocco, whereas the consumption of animal-based products followed an increasing trajectory [30]. However, contrary to our study findings, the consumption of fruits, vegetables and milk was found to increase in Portugal, based on national nutrition surveys [31]. This disparity in findings may reflect differences in the degree of nutrition transition, globalization of diet, and other economic and geopolitical factors that may impact the population's eating habits. Alternatively, the observed differences may be due to the use of different indicators for assessing the population's dietary trends. For instance, the study conducted in Portugal examined the percentage of consumers for each food group and how these percentages evolved over time, whereas in our study, the investigation of a temporal trend was based on a quantitative assessment of dietary intake [31].

Compared to our results, more favorable temporal trends in food consumption and nutrient intakes were reported from the US and some countries in Europe, particularly Northern European countries [4,6,32,33]. For instance, Rehm et al. showed, based on the National Health and Nutrition Examination Surveys (NHANES) (1999–2012), that the intakes of whole fruit, yogurt, dietary fiber and calcium, have increased in the US adult population, while the consumption of refined grains,

and added sugars followed a decreasing trend. In Europe, the French Individual and National Food Consumption Survey, which is based on 7-d dietary records, documented increases in overall fresh fruit consumption as well as vitamin C intakes in 18–79 years old adults (1998/1999 vs. 2006/2007), coupled with stable energy intakes [4]. In Switzerland, the Bus Santé Geneva Study (1999–2009), showed that intakes of energy, saturated fat and polyunsaturated fat decreased in participants aged 34–74 years, whereas intakes of carbohydrates (only women) increased, based on a semi-quantitative food frequency questionnaire [32]. Energy and fat intakes were also found to decrease among adults in the UK's National Diet and Nutrition Survey (assessed by 4-d dietary records) [33]. The observed favorable changes in these countries may be due to strong national nutrition policies and interventions, whereas the nutrition policy in Lebanon is still nonexistent [4,34,35]. It is however important to note that comparisons with other countries should be made with caution. For instance, the survey designs may differ as some studies adopted cohort designs [5]. Whereas others, as the case of our study, used repeated cross-sectional studies [4,33]. In addition, surveys conducted in different countries may have been performed at different time periods, among samples of varying age ranges, using different dietary assessment methods and nutrient databases [5].

Interestingly, some gender-based disparities in food consumption trends were noted. For instance, the decreases in total sugar intake and fresh fruit juice consumption were significant among women, but not men. This may reflect a higher level of health consciousness that is usually more commonly observed among women compared to men [11,36,37]. The decrease in milk intake and the increase in the consumption of chips and salty crackers were found to be significant only among younger adults, aged 20–39.9 years. A possible explanation may be that, in comparison with older adults, younger generations tend to have greater exposure to new and "fashionable" food products [38] and are more likely to adopt "modern" or "Westernized" eating habits [11]. Studies stemming from the 2008/2009 national survey have in fact documented a direct association between age and adherence to the traditional Lebanese dietary pattern, whereas adherence to the "Western Pattern" was more common in younger adults [11,17,36]. These observations are in many aspects reflective of the ongoing nutrition transition, a phenomenon that typically manifests itself in younger age groups [11,17,36].

The present study has several strengths. The study is based on individual food consumption surveys that were conducted on a nationally representative samples of adults. In both surveys (1997 and 2008/2009), the same dietary assessment methods and nutrient databases were used, and therefore the observed trends in food consumption and nutrient intakes cannot be attributed to variations in methods. In addition, both surveys have covered a wide age range, permitting the estimation of population-wide trends in food consumption and nutrient intakes over the study period. Instead of relying on self-reported data, the 24-HRs assessment was administered by trained nutritionists who received extensive training to reduce judgmental communication and minimize social desirability bias. However, the results of the study ought to be considered in light of the following limitations. First, the investigation of trends in food consumption and nutrient intakes was based on two distinct cross-sectional surveys with different sample populations. Second, it is important to mention that, in Lebanon, there exists no food composition database. Therefore, the USDA database of the Nutritionist Pro was used for dietary analysis including that of mixed dishes and endogenous recipes. Although this limitation could have affected the absolute estimates of dietary intake, it would have had limited effects on the trends in dietary intake, given that the same food composition database was used in both surveys. Third, the differences in dietary intake between the two surveys could be in part due to a birth cohort effect. Hence future research is needed to understand such an effect in order to formulate cohort-specific strategies to reverse the nutrition transition in the country [39]. Lastly, some of these differences, though statistically significant, were rather small in magnitude and hence their clinical implications could be limited.

## 5. Conclusions

This study showed that, between 1997 and 2008/2009, the diet of Lebanese adults has shifted towards higher intakes of energy and dietary fat, coupled with decreases in carbohydrates, micronutrient dietary density and the consumption of cardio-protective foods such as fruits. These dietary shifts were observed in both genders and across age, albeit there were some disparities between population groups. In many aspects, the observed changes in food consumption patterns are consistent with the dietary shifts that typically characterize the nutrition transition, and which have been associated with increases in the prevalence of diet-related NCDs such as obesity, diabetes and cancer [1,40]. In a country where NCD mortality is estimated at 84% of total deaths, the study findings laid the seeds for public health policies aimed at halting the erosion of traditional Lebanese food consumption patterns and the promotion of healthier diets. More recent studies on shifts in dietary intake and population's nutritional status are warranted to inform these policies. In this respect, food consumption should not be isolated from other activities that represent the "food event", such as production, processing, marketing, and social and cultural habits [41]. There is a need for a food system approach that focuses on the promotion of "healthier diets" as a concept that embraces health, quality, biodiversity, sustainability and cultural heritage [41,42].

**Supplementary Materials:** The following are available online at http://www.mdpi.com/2072-6643/11/8/1738/s1; Table S1: Mean daily intake of various food groups (g/day) (by age and sex) and their comparison between 1997 and 2008/2009, in the study population (*n* = 3581), aged ≥20 years, Lebanon; Table S2: Mean daily macronutrient intake (g/day) (by sex and age) and their comparison between 1997 and 2008/2009, in the study population (*n* = 3581), aged ≥20 years, Lebanon; Table S3: Mean daily micronutrient intake (g/day) (by sex and age) and their comparison between 1997 and 2008/2009, in the study population (*n* = 3581), aged ≥20 years, Lebanon.

**Author Contributions:** Conceptualization, F.N. and L.N.; methodology, N.H. and A.M.S.; formal analysis, J.J.A.; data curation, J.T.; writing—original draft preparation, F.N. and L.N.; writing—review and editing, F.N., L.N., and J.J.A.; supervision, F.N.; funding acquisition, F.H. All authors approved the final manuscript.

**Funding:** This research was funded by Food and Agricultural Organization of the UN (FAO), GCP/RNE/007/ITA.

**Conflicts of Interest:** The authors declare no conflict of interest.

## Appendix A

**Table A1.** Composition of each food group.

| Food Groups | Composition |
| --- | --- |
| Bread | All kinds of bread |
| Cereals and Cereal-Based Products | Cereals, rice, rice-based dishes, pasta, bulgur, pizza, pies |
| Legumes | All kinds of legumes, legume-based dishes |
| Starchy Vegetables | Potatoes (including potato-based dishes), pumpkin, sweet corn |
| Vegetables | Raw vegetables (including all kinds of salads and all kinds of vegetables), cooked vegetables, vegetable-based traditional dishes, canned vegetables (including heart of palm, carrots, tomato juice, artichoke) |
| Chips and Salty Crackers | Chips, pretzels, popcorn |
| Nuts and Seeds | All kinds of nuts and seeds (including peanuts, pine, pistachios, walnuts, flaxseed, pumpkin kernels, green almonds, coconut) |
| Dairy Products | |
| Milk | All kinds of milk (including infant formula) |
| Milk Derivatives | All kinds of cheese, yogurt, laban, yogurt-based dishes |
| Sweetened Milk | Milk-based puddings, frozen yogurt, fruit yogurt |

**Table A1.** *Cont.*

| Food Groups | Composition |
| --- | --- |
| Meat, Processed Meat, Poultry, Fish, Eggs | |
| 　Red Meat | Meat and meat organs |
| 　Processed Meat | Processed meat |
| 　Poultry | Poultry and poultry organs |
| 　Fish | All kinds of seafood |
| 　Eggs | Eggs |
| Fruits, Total | |
| 　Fruits | Fruits and dried fruits |
| 　Fresh Fruit Juices | All kinds of fresh fruits |
| Sweets and Added Sugars | |
| 　Sweets | Pastries, candies, biscuits, cakes, traditional sweets, ice cream |
| 　Added Sugars, Jams, Honey, Molasses | Added sugars, jams, honey, molasses |
| Sugar Sweetened Beverages | Sweetened juices, regular soft drinks |
| Hot Beverages (Coffee, Tea) | All kinds of coffee and tea |
| Alcoholic Beverages | Gin, rum, whiskey, arak, vodka, wine, beer |
| Added Fats and Oils | Olive oil, olives, avocado, sesame butter, all kinds of oils, mayonnaise, salad dressings, animal-based fat |
| Fast Food | All kinds of fast food |
| Miscellaneous | Pickles, soups, broth |

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
