# Peer review of "Differences in Dietary Intakes among Lebanese Adults over a Decade: Results from Two National Surveys 1997–2008/2009"

_nutrients, doi:10.3390/nu11081738_

Round 1

Reviewer 1 Report

The paper is written well, and it looks at a timely issue of dietary transition that is occurring in many countries and cultures including Lebanon.

I have the following suggestions/comments:

·     Abstract:Include the significance values (for example <0.001) when it stated, “A significant decrease was observed in the consumption of….”.  Or “significant increase in dietary energy (kcal/day) in the abstract. For example, significant increase, P< 0.05, in dietary energy (kcal/day).

·     Results:It would be good to add mean ±SD for age 

·     Could the relationship between some of the variables such as, age and total Kcal and fat   intake be tested when there is a claim that there is a difference between younger vs. older adults? 

·     Discussion/ results:The findings are indication that the intake of fat and Kcal increased, and intake of fruits and vegetables have decreased, as result, intake of some essential micronutrients is low in Lebanon compared with Portugal.  Could you explain this phenomenon in these two countries? Why we see this transitioning in one country but not the other? Are these shifts being due onlyto the demographic differences between the two surveys, 1997 and 2008-2009?  Or are there other factors such geopolitical struggles and other regional factors?   Some explanations are needed to clarify these transitions in the discussion section. 

The analysis of all the nutrients is comprehensive and could help to design and propose the factual intervention and policy. 

Author Response

Reviewer #1

1-      The title of the manuscript is a bit misleading as the authors do not quantify the change in any dietary measures. Perhaps the authors could report change (in the results) for some of the significant differences? If the authors presented or reported data on change, it might also help address one other issue, which is that some of the statistically significant differences between years were due to very small absolute changes. The limited clinical significance of small changes in % energy intake for some specific food groups should at least be addressed or mentioned.

Author’s response: We thank the reviewer his comment. Accordingly we revised the title to reflect better the analyses carried out in the study and its results. The revised title reads as follows ‘Differences in dietary intakes among Lebanese adults over a decade: results from two national surveys 1997-2008/2009’. We have also acknowledged in the discussion section of the manuscript that some of the differences observed in this study, though statistically significant, were rather small in magnitude and hence their clinical implications is limited. (Lines 419-420).

2-      Could the authors please indicate which analyses (either in the table footnotes or title of the corresponding tables) were adjusted for covariates.

Author’s response: As per the reviewer recommendation, adjustment for covariates was indicated as footnotes of the corresponding tables (footnote for tables 2, 3, and 4).

3-      Did the authors consider whether there were any possible birth cohort effects? It is difficult to evaluate given the way the age groups were broadly categorized.

Author’s response: Table 1 was revised to represent the age of the participants in both cohorts as means SD, in addition to the categories. As indicated by the reviewer, the possibility of a birth cohort effect cannot be ruled out and was added to the limitation section of the manuscript, as follows ‘Third, the differences in dietary intake between the two surveys could be in part due to a birth cohort effect. Hence future research is needed to understand such effect in order to formulate cohort-specific strategies to reverse the nutrition transition in the country’ (Lines416-419)

4-      Given that these data are now 10 years old, their relevance with respect to nutrition policies and interventions that would presumably be conducted in the present day are unclear. It would be helpful to know whether additional follow-up surveys are planned

Author’s response: We agree with the reviewer’s comment. Accordingly the text of the conclusion section of the manuscript was revised as follows ‘In a country where NCD mortality is estimated at 84% of total deaths, the study findings laid the seeds for public health policies aimed at halting the erosion of traditional Lebanese food consumption patterns and the promotion of healthier diets. More recent studies on shifts in dietary intake and population’s nutritional status are warranted to inform these policies.’ (Lines 429-434)

5-      Acronyms should be spelled out the first time they are used.

Author’s response: As per the reviewer’s comments, the acronyms have been revised and spelled out.

Reviewer 2 Report

The authors report here a descriptive analysis of dietary intake data from two national cross-sectional surveys conducted during different time periods.  The authors report significant decreases in the consumption of specific healthy food groups, and increases in total energy, % energy from fat as well as other specific less healthy food groups, documenting the nutrition transition of this country. This paper is very well written and motivated and uses appropriate statistical methods.  Relatively minor comments are summarized below. 

·        The title of the manuscript is a bit misleading as the authors do not quantify the change in any dietary measures.  Perhaps the authors could report change (in the results) for some of the significant differences?  If the authors presented or reported data on change, it might also help address one other issue, which is that some of the statistically significant differences between years were due to very small absolute changes.  The limited clinical significance of small changes in % energy intake for some specific food groups should at least be addressed or mentioned.

·        Could the authors please indicate which analyses (either in the table footnotes or title of the corresponding tables) were adjusted for covariates.

·        Did the authors consider whether there were any possible birth cohort effects?  It is difficult to evaluate given the way the age groups were broadly categorized.

·        Given that these data are now 10 years old, their relevance with respect to nutrition policies and interventions that would presumably be conducted in the present day are unclear.  It would be helpful to know whether additional follow-up surveys are planned.

·        Acronyms should be spelled out the first time they are used.

Author Response

1-      The title of the manuscript is a bit misleading as the authors do not quantify the change in any dietary measures. Perhaps the authors could report change (in the results) for some of the significant differences? If the authors presented or reported data on change, it might also help address one other issue, which is that some of the statistically significant differences between years were due to very small absolute changes. The limited clinical significance of small changes in % energy intake for some specific food groups should at least be addressed or mentioned.

Author’s response: We thank the reviewer his comment. Accordingly we revised the title to reflect better the analyses carried out in the study and its results. The revised title reads as follows ‘Differences in dietary intakes among Lebanese adults over a decade: results from two national surveys 1997-2008/2009’. We have also acknowledged in the discussion section of the manuscript that some of the differences observed in this study, though statistically significant, were rather small in magnitude and hence their clinical implications is limited. (Lines 419-420).

2-      Could the authors please indicate which analyses (either in the table footnotes or title of the corresponding tables) were adjusted for covariates.

Author’s response: As per the reviewer recommendation, adjustment for covariates was indicated as footnotes of the corresponding tables (footnote for tables 2, 3, and 4).

3-      Did the authors consider whether there were any possible birth cohort effects? It is difficult to evaluate given the way the age groups were broadly categorized.

Author’s response: Table 1 was revised to represent the age of the participants in both cohorts as means SD, in addition to the categories. As indicated by the reviewer, the possibility of a birth cohort effect cannot be ruled out and was added to the limitation section of the manuscript, as follows ‘Third, the differences in dietary intake between the two surveys could be in part due to a birth cohort effect. Hence future research is needed to understand such effect in order to formulate cohort-specific strategies to reverse the nutrition transition in the country’ (Lines416-419)

4-      Given that these data are now 10 years old, their relevance with respect to nutrition policies and interventions that would presumably be conducted in the present day are unclear. It would be helpful to know whether additional follow-up surveys are planned

Author’s response: We agree with the reviewer’s comment. Accordingly the text of the conclusion section of the manuscript was revised as follows ‘In a country where NCD mortality is estimated at 84% of total deaths, the study findings laid the seeds for public health policies aimed at halting the erosion of traditional Lebanese food consumption patterns and the promotion of healthier diets. More recent studies on shifts in dietary intake and population’s nutritional status are warranted to inform these policies.’ (Lines 429-434)

5-      Acronyms should be spelled out the first time they are used.

Author’s response: As per the reviewer’s comments, the acronyms have been revised and spelled out.

Reviewer 3 Report

This is a study of to characterize the nutrition transition in the Eastern Mediterranean Region, based on individual national nutrition surveys. We recognized their food change to high protein, fat and low in CHO. May try to connect to health status in the future.

Author Response

1-      This is a study to characterize the nutrition transition in the Eastern Mediterranean Region, based on individual national nutrition surveys. We recognized their food change to high protein, fat and low in CHO. May try to connect to health status in the future

Author’s response: We thank the reviewer for this comment. We have amended the text to highlight the need for future studies that investigate dietary shifts and nutritional status in Lebanon, as follows:

“More recent studies on shifts in dietary intake and population’s nutritional status are warranted to inform these policies”. (Lines 432-434).
